# Changes in parental smoking during pregnancy and risks of adverse birth outcomes and childhood overweight in Europe and North America: An individual participant data meta-analysis of 229,000 singleton births

Elise M. Philips[1,2], Susana Santos[1,2], Leonardo Trasande[3,4,5,6,7], Juan J. Aurrekoetxea[8,9,10], Henrique Barros[11,12], Andrea von Berg[13], Anna Bergström[14,15], Philippa K. Bird[16], Sonia Brescianini[17], Carol Ní Chaoimh[18,19], Marie-Aline Charles[20], Leda Chatzi[21], Cécile Chevrier[22], George P. Chrousos[23], Nathalie Costet[22], Rachel Criswell[24,25], Sarah Crozier[26], Merete Eggesbø[27], Maria Pia Fantini[28], Sara Farchi[29], Francesco Forastiere[29], Marleen M. H. J. van Gelder[30,31], Vagelis Georgiu[32], Keith M. Godfrey[26,33], Davide Gori[28], Wojciech Hanke[34], Barbara Heude[20], Daniel Hryhorczuk[35], Carmen Iñiguez[10,36], Hazel Inskip[26,33], Anne M. Karvonen[37], Louise C. Kenny[19,38], Inger Kull[39,40], Debbie A. Lawlor[41,42], Irina Lehmann[43], Per Magnus[44], Yannis Manios[45], Erik Melén[14,39,40], Monique Mommers[46], Camilla S. Morgen[47,48], George Moschonis[49], Deirdre Murray[19,50], Ellen A. Nohr[51], Anne-Marie Nybo Andersen[48], Emily Oken[52], Adriëtte J. J. M. Oostvogels[53], Eleni Papadopoulou[54], Juha Pekkanen[37,55], Costanza Pizzi[56], Kinga Polanska[34], Daniela Porta[29], Lorenzo Richiardi[56], Sheryl L. Rifas-Shiman[52], Nel Roeleveld[30], Franca Rusconi[57], Ana C. Santos[11,12], Thorkild I. A. Sørensen[48,58], Marie Standl[59], Camilla Stoltenberg[60,61], Jordi Sunyer[10,62,63], Elisabeth Thiering[59,64], Carel Thijs[46], Maties Torrent[65], Tanja G. M. Vrijkotte[53], John Wright[66], Oleksandr Zvinchuk[67], Romy Gaillard[1,2], Vincent W. V. Jaddoe[1,2]*

1 The Generation R Study Group, Erasmus University Medical Center, Rotterdam, the Netherlands, 2 Department of Pediatrics, Sophia Children's Hospital, Erasmus University Medical Center, Rotterdam, the Netherlands, 3 Department of Pediatrics, New York University School of Medicine, New York City, New York, United States of America, 4 Department of Environmental Medicine, New York University School of Medicine, New York City, New York, United States of America, 5 Department of Population Health, New York University School of Medicine, New York City, New York, United States of America, 6 New York Wagner School of Public Service, New York City, New York, United States of America, 7 New York University College of Global Public Health, New York City, New York, United States of America, 8 Subdirección de Salud Pública Gipuzkoa, San Sebastián, Spain, 9 Instituto de Investigación Sanitaria BIODONOSTIA, San Sebastián, Spain, 10 CIBER Epidemiología y Salud Pública (CIBERESP), Madrid, Spain, 11 EPIUnit—Instituto de Saúde Pública, Universidade do Porto, Porto, Portugal, 12 Department of Public Health and Forensic Sciences and Medical Education, Unit of Clinical Epidemiology, Predictive Medicine and Public Health, University of Porto Medical School, Porto, Portugal, 13 Research Institute, Department of Pediatrics, Marien-Hospital Wesel, Wesel, Germany, 14 Institute of Environmental Medicine, Karolinska Institutet, Stockholm, Sweden, 15 Centre for Occupational and Environmental Medicine Stockholm County Council, Stockholm, Sweden, 16 Born in Bradford, Bradford Institute for Health Research, Bradford Teaching Hospitals NHS Foundation Trust, Bradford, United Kingdom, 17 Centre for Behavioural Science and Mental Health, Istituto Superiore di Sanità, Rome, Italy, 18 Cork Centre for Vitamin D and Nutrition Research, School of Food and Nutritional Sciences, University College Cork, Cork, Ireland, 19 Irish Centre for Fetal and Neonatal Translational Research, Cork University Maternity Hospital, University College Cork, Cork, Ireland, 20 Université de Paris, CRESS, INSERM, INRA, Paris, France, 21 Department of Preventive Medicine, University of Southern California, Los Angeles, United States of America, 22 Univ Rennes, Inserm, EHESP, Irset (Institut de recherche en santé, environnement et travail)–UMR_S 1085, Rennes, France, 23 First Department of Pediatrics, National and Kapodistrian University of Athens Medical School, Aghia Sophia Children's Hospital, Athens, Greece, 24 Department of Environmental Exposure and Epidemiology, Norwegian Institute of Public Health, Oslo, Norway, 25 Maine-Dartmouth Family Medicine Residency, Augusta, Maine, United States of America, 26 MRC Lifecourse Epidemiology Unit, University of Southampton,

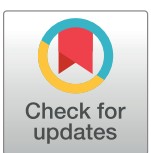

**Data Availability Statement:** Data from 28 different cohorts with different data publishing policies were used. The datasets generated and analyzed for this study are available upon request to the executive committees of the individual cohorts. Contacts for data requests for each cohort are listed in S9 Table.

**Funding:** This collaborative project received funding from the European Union's Horizon 2020 research and innovation programme (Grant Agreement No. 733206 LifeCycle). EMP and LT were supported by grant R01ES022972 from the National Institutes of Health, USA. LC was supported by the National Institute for Environmental Health Sciences: P30ES007048, R21ES029681, R01ES029944, R01ES030364, R21ES028903. DAL works in a unit that receives UK MRC funding (MC_UU_12013/5) and is an NIHR senior investigator (NF-SI-0611-10196). ACS holds an FCT Investigator contract IF/01060/2015. RG received funding from the Dutch Heart Foundation (grant number 2017T013), the Dutch Diabetes Foundation (grant number 2017.81.002) and the Netherlands Organization for Health Research and Development (ZonMW, 543003109). VWVJ received grant from the European Research Council (Consolidator Grant, ERC-2014-CoG-648916). Cohort-specific sources of funding are listed in S2 Text. The funders had no role in study design, data collection and analysis, decision to publish, or preparation of the manuscript.

**Competing interests:** I have read the journal's policy and the authors of this manuscript have the following competing interests: AvB has received reimbursement for speaking at symposia sponsored by Nestlé and Mead Johnson, who partly financially supported the 15-year follow-up examination of the GINIplus study. KMG has received reimbursement for speaking at conferences sponsored by companies selling nutritional products and is part of an academic consortium that has received research funding from Abbott Nutrition, Nestec, and Danone. DAL has received support from Roche Diagnostics and Medtronic in relation to biomarker research that is not related to the research presented in this paper. The other authors have declared that no competing interests exist.

**Abbreviations:** BMI, body mass index; CI, confidence interval; IV, instrumental variable; OR, odds ratio; PRISMA, Preferred Reporting Items for Systematic Reviews and Meta-Analyses; SDS, standard deviation score; SGA, small size for gestational age; WHO, World Health Organization..

Southampton, United Kingdom, **27** Department of Exposure and Environmental Epidemiology, Norwegian Institute of Public Health, Oslo, Norway, **28** The Department of Biomedical and Neuromotor Sciences, University of Bologna, Bologna, Italy, **29** Department of Epidemiology, Lazio Regional Health Service, Rome, Italy, **30** Department for Health Evidence, Radboud Institute for Health Sciences, Radboud University Medical Center, Nijmegen, the Netherlands, **31** Radboud REshape Innovation Center, Radboud University Medical Center, Nijmegen, the Netherlands, **32** Department of Social Medicine, University of Crete, Heraklion, Greece, **33** NIHR Southampton Biomedical Research Centre, University of Southampton and University Hospital Southampton NHS Foundation Trust, Southampton, United Kingdom, **34** Department of Environmental Epidemiology, Nofer Institute of Occupational Medicine, Lodz, Poland, **35** Center for Global Health, University of Illinois College of Medicine, Chicago, Illinois, United States of America, **36** Department of Statistics and Computational Research, Universitat de València, València, Spain, **37** Department of Health Security, Finnish Institute for Health and Welfare, Kuopio, Finland, **38** Department of Obstetrics and Gynaecology, Cork University Maternity Hospital, Cork, Ireland, **39** Department of Clinical Science and Education, Södersjukhuset, Karolinska Institutet, Stockholm, Sweden, **40** Sachs' Children and Youth Hospital, Stockholm, Sweden, **41** MRC Integrative Epidemiology Unit at the University of Bristol, Oakfield House, Oakfield Grove, Bristol, United Kingdom, **42** Population Health Science, Bristol Medical School, University of Bristol, Bristol, United Kingdom, **43** Department of Environmental Immunology/Core Facility Studies, Helmholtz Centre for Environmental Research–UFZ, Leipzig, Germany, **44** Division of Health Data and Digitalization, Norwegian Institute of Public Health, Oslo, Norway, **45** Department of Nutrition and Dietetics, School of Health Science and Education, Harokopio University, Athens, Greece, **46** Department of Epidemiology, Care and Public Health Research Institute, Maastricht University Maastricht, the Netherlands, **47** National Institute of Public Health, University of Southern Denmark, Copenhagen, Denmark, **48** Department of Public Health, Section of Epidemiology, Faculty of Health and Medical Sciences, University of Copenhagen, Copenhagen, Denmark, **49** Department of Dietetics, Nutrition and Sport, School of Allied Health, Human Services and Sport, La Trobe University, Melbourne, Australia, **50** Paediatrics & Child Health, University College Cork, Cork, Ireland, **51** Research Unit for Gynaecology and Obstetrics, Institute for Clinical Research, University of Southern Denmark, Denmark, **52** Department of Population Medicine, Harvard Medical School, Harvard Pilgrim Health Care Institute, Boston, Massachusetts, United States of America, **53** Department of Public Health, Amsterdam Public Health Research Institute, Academic Medical Center, Amsterdam, the Netherlands, **54** Department of Environmental Exposures and Epidemiology, Domain of Infection Control and Environmental Health, Norwegian Institute of Public Health, Oslo, Norway, **55** Department of Public Health, University of Helsinki, Helsinki, Finland, **56** Department of Medical Sciences, University of Turin, Turin, Italy, **57** Unit of Epidemiology, "Anna Meyer" Children's University Hospital, Florence, Italy, **58** The Novo Nordisk Foundation Center for Basic Metabolic Research, Section on Metabolic Genetics, Faculty of Health and Medical Sciences, University of Copenhagen, Copenhagen, Denmark, **59** Institute of Epidemiology, Helmholtz Zentrum München-German Research Center for Environmental Health, Neuherberg, Germany, **60** Norwegian Institute of Public Health, Oslo, Norway, **61** Department of Global Public Health and Primary Care, University of Bergen, Bergen, Norway, **62** ISGlobal, Institute for Global Health, Barcelona, Spain, **63** Universitat Pompeu Fabra (UPF), Barcelona, Spain, **64** Dr. von Hauner Children's Hospital, Ludwig-Maximilians-University Munich, Munich, Germany, **65** Ib-salut, Area de Salut de Menorca, Palma, Spain, **66** Bradford Institute for Health Research, Temple Bank House, Bradford Royal Infirmary, Duckworth Lane, Bradford, United Kingdom, **67** Department of Medical and Social Problems of Family Health, Institute of Pediatrics, Obstetrics and Gynecology, Kyiv, Ukraine

☙ These authors contributed equally to this work.
* v.jaddoe@erasmusmc.nl

## Abstract

### Background

Fetal smoke exposure is a common and key avoidable risk factor for birth complications and seems to influence later risk of overweight. It is unclear whether this increased risk is also present if mothers smoke during the first trimester only or reduce the number of cigarettes during pregnancy, or when only fathers smoke. We aimed to assess the associations of parental smoking during pregnancy, specifically of quitting or reducing smoking and maternal and paternal smoking combined, with preterm birth, small size for gestational age, and childhood overweight.

## Methods and findings

We performed an individual participant data meta-analysis among 229,158 families from 28 pregnancy/birth cohorts from Europe and North America. All 28 cohorts had information on maternal smoking, and 16 also had information on paternal smoking. In total, 22 cohorts were population-based, with birth years ranging from 1991 to 2015. The mothers' median age was 30.0 years, and most mothers were medium or highly educated. We used multilevel binary logistic regression models adjusted for maternal and paternal sociodemographic and lifestyle-related characteristics. Compared with nonsmoking mothers, maternal first trimester smoking only was not associated with adverse birth outcomes but was associated with a higher risk of childhood overweight (odds ratio [OR] 1.17 [95% CI 1.02–1.35], $P$ value = 0.030). Children from mothers who continued smoking during pregnancy had higher risks of preterm birth (OR 1.08 [95% CI 1.02–1.15], $P$ value = 0.012), small size for gestational age (OR 2.15 [95% CI 2.07–2.23], $P$ value < 0.001), and childhood overweight (OR 1.42 [95% CI 1.35–1.48], $P$ value < 0.001). Mothers who reduced the number of cigarettes between the first and third trimester, without quitting, still had a higher risk of small size for gestational age. However, the corresponding risk estimates were smaller than for women who continued the same amount of cigarettes throughout pregnancy (OR 1.89 [95% CI 1.52–2.34] instead of OR 2.20 [95% CI 2.02–2.42] when reducing from 5–9 to ≤4 cigarettes/day; OR 2.79 [95% CI 2.39–3.25] and OR 1.93 [95% CI 1.46–2.57] instead of OR 2.95 [95% CI 2.75–3.15] when reducing from ≥10 to 5–9 and ≤4 cigarettes/day, respectively [$P$ values < 0.001]). Reducing the number of cigarettes during pregnancy did not affect the risks of preterm birth and childhood overweight. Among nonsmoking mothers, paternal smoking was associated with childhood overweight (OR 1.21 [95% CI 1.16–1.27], $P$ value < 0.001) but not with adverse birth outcomes. Limitations of this study include the self-report of parental smoking information and the possibility of residual confounding. As this study only included participants from Europe and North America, results need to be carefully interpreted regarding other populations.

## Conclusions

We observed that as compared to nonsmoking during pregnancy, quitting smoking in the first trimester is associated with the same risk of preterm birth and small size for gestational age, but with a higher risk of childhood overweight. Reducing the number of cigarettes, without quitting, has limited beneficial effects. Paternal smoking seems to be associated, independently of maternal smoking, with the risk of childhood overweight. Population strategies should focus on parental smoking prevention before or at the start, rather than during, pregnancy.

### Author summary

#### Why was this study done?

- Maternal smoking during pregnancy is an important risk factor for various birth complications and childhood overweight.

- It is not clear whether this increased risk is also present if mothers smoke during the first trimester only or reduce the number of cigarettes during pregnancy.

- The associations of paternal smoking with birth and childhood outcomes also remain unknown.

## What did the researchers do and find?

- We conducted an individual participant data meta-analysis using data from 229,158 families from 28 pregnancy and birth cohorts from Europe and North America to assess the associations of parental smoking during pregnancy, specifically of quitting or reducing smoking and maternal and paternal smoking combined, with preterm birth, small size for gestational age, and childhood overweight.

- We observed that smoking in the first trimester only did not increase the risk of preterm birth and small size for gestational age but was associated with a higher risk of childhood overweight, as compared to nonsmoking. Reducing the number of cigarettes during pregnancy, without quitting, was still associated with higher risks of these adverse outcomes.

- Paternal smoking seems to be associated, independently of maternal smoking, with the risks of childhood overweight.

## What do these findings mean?

- Population strategies should focus on parental smoking prevention before or at the start of, rather than during, pregnancy.

- Future studies are needed to assess the specific associations of smoking in the preconception and childhood periods with offspring outcomes.

## Introduction

One in five women of reproductive age are expected to be tobacco users by 2025 [1]. Although strategies to prevent smoking are globally implemented, up to 25% of women in Western countries smoke during pregnancy [2]. This is a major public health concern, particularly since smoking during pregnancy not only affects women's own health but is also associated with adverse birth and offspring outcomes, such as preterm birth, low birth weight, and childhood overweight [3–13]. Preterm birth and low birth weight are major causes of perinatal morbidity and mortality, and childhood overweight is related to a higher risk of cardiovascular disease, premature death, and disability in adulthood [14–16].

A vast number of studies observed consistent associations of continued maternal smoking during pregnancy with increased risks of preterm birth, low birth weight, and childhood overweight [7,10,11]. However, evidence on critical windows of vulnerability to maternal smoking and changes in smoking behavior during pregnancy remain inconclusive, potentially reflecting between-study heterogeneity of outcome measures and small study sample sizes. Previous studies focusing on maternal smoking in first trimester of pregnancy only consistently showed no associations with preterm birth but showed conflicting results for the risks of low birth weight and childhood overweight [8,9,17–21]. Also, the associations of paternal smoking during pregnancy with preterm birth, low birth weight, and childhood overweight have been

scarcely studied and remain unclear [20,22,23]. Paternal smoking might affect offspring outcomes through direct gamete or passive smoking intrauterine effects. However, comparisons of maternal and paternal smoking associations can also be used to disentangle direct uterine programming effects and confounding by shared or family-based lifestyle or socioeconomic variables. To our knowledge, no large sample size studies assessed the associations of maternal smoking during first trimester only, of reducing the number of cigarettes during pregnancy, or of paternal smoking only with birth and childhood outcomes.

We conducted an individual participant data meta-analysis among 229,158 singleton births from 28 pregnancy and birth cohort studies in Europe and North America to assess the associations of parental smoking during pregnancy with preterm birth, small size for gestational age (SGA), and childhood overweight. We were specifically interested in the associations of quitting or reducing smoking during pregnancy and of combined maternal and paternal smoking patterns with birth and offspring outcomes.

## Methods

### Inclusion criteria and participating cohorts

This study was part of an international LifeCycle Project (https://lifecycle-project.eu) collaboration on maternal obesity and childhood outcomes [24–28]. Pregnancy and birth cohort studies were eligible for inclusion if they included mothers with singleton live-born children who were born from 1989 onwards, had information available on maternal prepregnancy/early-pregnancy body mass index (BMI), and had at least one offspring measurement (birth weight or childhood BMI). We identified eligible cohorts from existing collaborations on childhood health (EarlyNutrition Project, CHICOS Project, www.birthcohorts.net assessed until July 2014). Fifty cohorts from Europe, North America, and Oceania were identified and invited, of which 39 cohorts agreed to participate. The cohorts were approved by their local institutional review boards, and written informed consent from all participants or parents was obtained. Eleven cohorts were excluded from the current analysis because there was no information on maternal smoking patterns or only nonsmoking mothers in their cohort. In total, 28 cohorts comprising data on 229,158 singleton births were included (Fig 1). Twenty-two of the 28 cohorts defined themselves as regionally or nationally based studies, four as hospital-based (Co.N.ER, EDEN, GASPII, LUKAS), one as internet users–based (NINFEA), and one as studying selected populations (FCOU). The plan for analyses given to the cohorts when inviting them to participate in this paper from the LifeCycle Project collaboration is provided in S1 Text. Based on data availability and additional research questions, it was decided among the collaborators to refine the existing questions and to extend the project with additional questions to be addressed. Analyses that were not in the original plan are marked in S1 Text. Associations of smoking with early- and late-childhood BMI were excluded because of low numbers. All cohorts provided written informed consent for using their data. Anonymized datasets were stored on a single central secured data server with access for the main analysts (EP, SS) only. This study is reported according to the Preferred Reporting Items for Systematic Reviews and Meta-Analyses (PRISMA) guideline (S1 PRISMA Checklist).

### Parental tobacco smoking

Parental smoking information was obtained by questionnaires (cohort-specific information in S1 Table). We used trimester-specific maternal smoking information to categorize smoking during pregnancy in three groups (nonsmoking; first-trimester-only smoking; continued smoking [as being any second or third trimester smoking]). Trimester-specific maternal smoking information was categorized into nonsmoking, ≤4 cigarettes/day, 5–9 cigarettes/day,

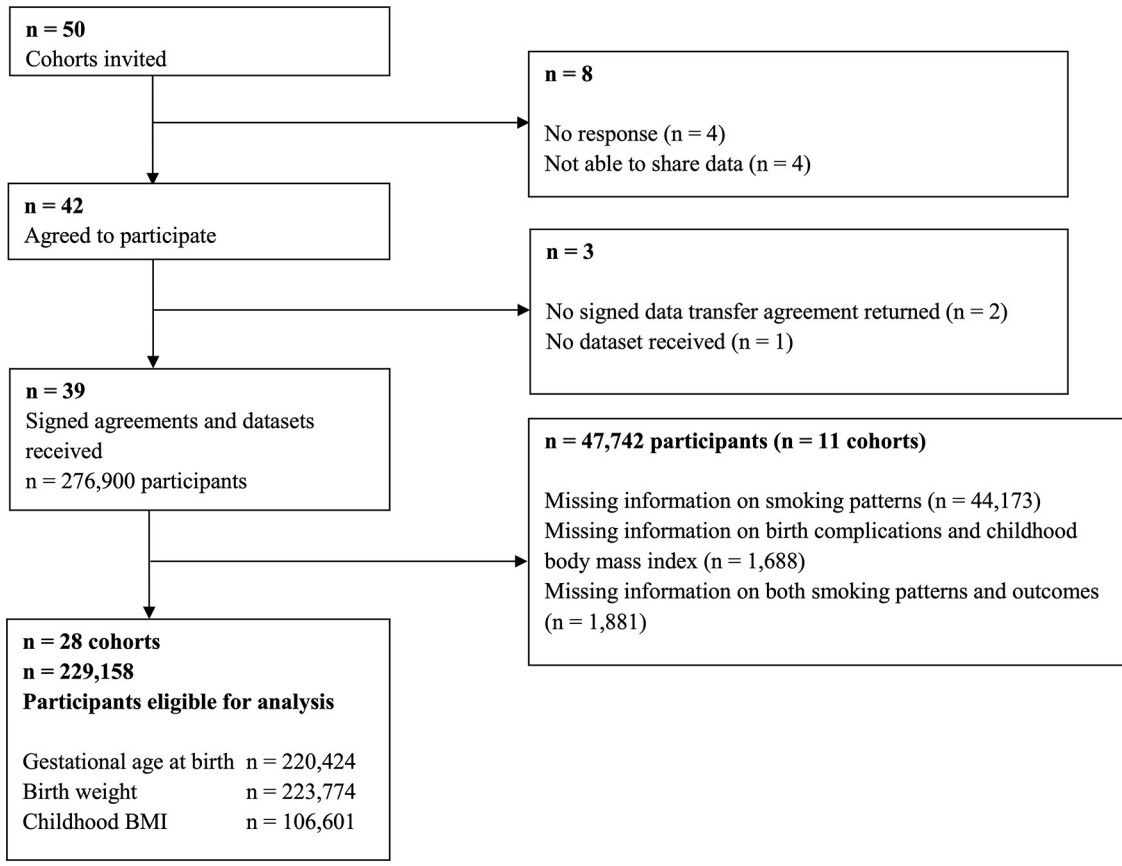

**Fig 1. Flowchart of the cohorts and participants.** BMI, body mass index.

and ≥10 cigarettes/day. We combined the information about maternal smoking in first and third trimester to examine the change in smoking behavior. Information on paternal non-smoking/smoking was used. To explore the combined effects of maternal and paternal smoking, we combined the maternal and paternal smoking information into six categories: maternal and paternal nonsmoking (used as reference category); maternal nonsmoking and paternal smoking; maternal first-trimester-only smoking and paternal nonsmoking; maternal first-trimester-only smoking and paternal smoking; maternal continued smoking and paternal nonsmoking; and maternal continued smoking and paternal smoking.

## Birth complications and childhood overweight

Information on gestational age at birth, birth weight, and childhood weight and height was measured, derived from clinical records, or reported (cohort-specific information in **S1 Table**). Preterm birth was defined as <37 weeks of gestation, and full-term birth (≥37 weeks) was used as the reference group in the analyses [29]. We created sex- and gestational age–adjusted birth weight standard deviation scores (SDSs) based on a North European reference chart [30]. SGA at birth was defined per cohort as sex- and gestational age–adjusted birth weight below the 10th percentile. The reference group used in the analyses comprises children born at appropriate and large size for gestational age (i.e., cohort-specific sex- and gestational age–adjusted birth weight above the 10th percentile). BMI measurements in mid-childhood (≥5 to <10 years) were used. If there were multiple measurements of a child available within

the age interval, we used the measurement at the highest age. We created sex- and age-adjusted SDSs of childhood BMI using World Health Organization (WHO) reference growth charts (Growth Analyzer 4.0, Dutch Growth Research Foundation) [31,32]. Childhood normal weight, overweight, and obesity were defined using WHO cutoffs [31,32]. For the analyses, we combined the overweight and obesity group, hereafter referred to as the overweight group. Normal weight was used as the reference group in childhood overweight analyses.

## Covariates

Information on covariates was mostly assessed using questionnaires. Most covariates were provided by cohorts as categorical variables: child's sex, maternal educational level (low, medium, high), parity (nulliparous, multiparous), and alcohol consumption during pregnancy (yes, no). To allow handling of missing data, continuous covariates were categorized: maternal age (defined on the basis of data availability: <25.0 years, 25.0–29.9 years, 30.0–34.9 years, and ≥35.0 years) and prepregnancy or early-pregnancy maternal and paternal BMI (underweight [<18.5 kg/m$^2$], normal weight [18.5–24.9 kg/m$^2$], overweight [25.0–29.9 kg/m$^2$], and obesity [≥30.0 kg/m$^2$]). Maternal ethnicity was not included, since most cohorts were largely of European descent and there was a high percentage of missing data. Covariates per cohort are described in **S2 Table**.

## Statistical analysis

We conducted 1-stage meta-analyses, in which we analyzed individual participant data from all cohorts simultaneously in binary logistic multilevel mixed-effects models, accounting for clustering of participants within cohorts [33]. First, we examined the associations of maternal smoking (across different trimesters; dose-response) with the risks of preterm birth, SGA, and childhood overweight. When examining the dose-response effects of first trimester maternal smoking, mothers who continued smoking were excluded from the analysis. Second, we used similar models to investigate the associations of change in maternal smoking behavior from first to third trimester with the risks of preterm birth, SGA, and childhood overweight. Finally, we used similar models to investigate the combined associations of both maternal and paternal smoking with the risks of these outcomes. We assessed whether the risk estimates between categories statistically differed using the formula $Z = \frac{\beta1-\beta2}{\sqrt{(SE\beta1)^2+(SE\beta2)^2}}$ [34]. We adjusted all analyses focused on maternal smoking for maternal age, educational level, parity, prepregnancy or early-pregnancy BMI, alcohol consumption during pregnancy, and paternal smoking. We adjusted all analyses focused on combined maternal and paternal smoking for the same covariates and paternal BMI. As sensitivity analyses, we repeated all models for gestational age at birth, sex- and gestational age–adjusted birth weight SDSs, and childhood sex- and age-adjusted BMI SDSs. Also, we conducted two-stage random-effects meta-analyses for the core associations and tested for heterogeneity between the cohorts estimates with the I$^2$ test [33,35]. To express the uncertainty associated with I$^2$ estimates, we calculated the corresponding 95% confidence intervals (CIs) [36]. All covariates were categorized and missing values were added as an additional group to prevent exclusion of noncomplete cases. If information on a covariate was available for less than 50% of the cohort sample used for each analysis, available information was not used and the corresponding data for that full cohort sample were assigned to the missing category. We conducted a sensitivity analysis with complete cases only. Also, to explore the influence on our results of using maternal age and BMI as categorical covariates, we repeated the complete cases' analysis using these covariates continuously. The statistical analyses were performed using the Statistical Package of Social Sciences version 24.0 for

Windows (SPSS, Chicago, IL, United States of America) and Review Manager (RevMan) version 5.3 of the Cochrane Collaboration (The Nordic Cochrane Centre, Copenhagen, Denmark).

## Results

### Participants' characteristics

Information about the main characteristics per cohort is given in **Table 1**. Overall, 14.4% (range 5.5–26.8) of mothers and 27.5% (range 16.9–83.8) of fathers smoked during pregnancy. Children were born at a median gestational age of 40.0 weeks (95% range 35.7–42.3) and a median birth weight of 3,530 grams (95% range 2,390–4,580). In total, 4.7% of children were born preterm, 10.0% were SGA at birth, and 20% were in the overweight group. Additional information about maternal smoking is given in **S3 Table**.

### Changes in maternal smoking habits during pregnancy and the risks of preterm birth, SGA, and childhood overweight

**Table 2** shows that maternal first trimester smoking only was not associated with adverse birth outcomes but was associated with higher risks of childhood overweight (odds ratio [OR] 1.17 (95% CI 1.02–1.35), $P$ value = 0.030). Compared with children from mothers who did not smoke during pregnancy, those from mothers who continued smoking had higher risks of preterm birth (OR 1.08 [1.02–1.15], $P$ value = 0.012), SGA (OR 2.15 [2.07–2.33], $P$ value < 0.001), and childhood overweight (OR 1.42 [1.35–1.48], $P$ value < 0.001). We observed dose-response relationships for third trimester smoking starting at ≤4 cigarettes/day. We observed similar results when we used the continuous outcomes, except for the association of first-trimester-only smoking with childhood BMI SDS, which was in the same direction but no longer significant (S4 Table). We observed similar results when using two-stage random-effects models (**Figs 2, 3 and 4**). We observed low to moderate heterogeneity between the cohorts' estimates ($I^2$ estimates range from 0% to 47%; corresponding CIs are presented in the footnotes of Figs 2, 3 and 4). Only the cohort-specific results for the associations of maternal continued smoking with SGA showed high heterogeneity between estimates ($I^2$ 75% [95% CI 56%–86%]). Almost all cohorts were included in the analyses for continued smoking, whereas only roughly half had information on first-trimester-only smoking. When restricting the two-stage continued smoking models to the cohorts also with information on first-trimester-only smoking, we observed a lower heterogeneity between estimates ($I^2$ 23% [95% CI 0%–65%]), but the pooled risk estimate remained similar (S1 Fig).

**Table 3** shows that, compared with mothers who did not smoke during pregnancy, mothers who quit smoking from first to third trimester had similar risks of delivering SGA infants. Reducing the number of cigarettes, without quitting, from first to third trimester lowered the risks of delivering SGA infants, but risks were still higher compared with those of nonsmoking mothers (OR 1.89 [1.52–2.34] when reducing from 5–9 to ≤4 cigarettes/day; 2.79 [2.39–3.25] and 1.93 [1.46–2.57] when reducing from ≥10 to 5–9 and ≤ 4 cigarettes/day, respectively [all $P$ values < 0.001]). Mothers who increased the number of cigarettes from first to third trimester increased their risks of delivering SGA infants (OR 2.43 [2.05–2.89] and 2.47 [1.71–3.58] when increasing from ≤4 to 5–9 and ≥10 cigarettes/day, respectively; and 2.70 [2.35–3.10] when increasing from 5–9 to ≥10 cigarettes/day [all $P$ value < 0.001]). Changes in maternal smoking from first to third trimester did not influence the risks of preterm birth and childhood overweight. Similar results were observed when assessing the associations of the changes in maternal smoking during pregnancy with the continuous outcomes (S5 Table).

**Table 1. Characteristics of the participating pregnancy and birth cohorts (*n* = 229,158).**

| Cohort name, number of participants, birth years (country) | Maternal smoking | | | Paternal smoking | | Birth outcomes | | | | Childhood BMI | | |
|---|---|---|---|---|---|---|---|---|---|---|---|---|
| | No | First trimester only | Continued | No | Yes | Gestational age at birth (weeks) | Preterm birth | Birth weight (g) | Small size for gestational age at birth | Age (months) | BMI (SDS) | Overweight |
| ABCD, *n* = 7,324, 2003–2004 (the Netherlands) | 6,571 (89.7) | NA | 753 (10.3) | NA | NA | 40.0 (35.0–42.0) | 385 (5.3) | 3,460 (2,270–4,500) | 732 (10.1) | 68.1 (61.6–82.1) | 0.09 (−1.69 to 2.29) | 706 (16.6) |
| ALSPAC, *n* = 12,148, 1991–1992 (United Kingdom) | 9,581 (78.9) | NA | 2,567 (21.1) | 7,397 (63.2) | 4,301 (36.8) | 40.0 (35.0–42.0) | 650 (5.4) | 3,440 (2,240–4,420) | 1,190 (10.0) | 115.0 (88.0–119.0) | 0.24 (−1.61 to 2.66) | 1,960 (26.3) |
| BAMSE, *n* = 4,057, 1994–1996 (Sweden) | 3,533 (87.1) | 72 (1.8) | 452 (11.1) | 2,756 (83.1) | 560 (16.9) | 40.0 (35.0–42.0) | 212 (5.3) | 3,545 (2,334–4,550) | 396 (9.9) | 101.0 (89.0–109.0) | 0.52 (−1.20 to 2.63) | 814 (31.2) |
| BIB, *n* = 1,641, 2007–2010 (UK) | 1,398 (85.2) | NA | 243 (14.8) | NA | NA | 39.7 (35.3–41.9) | 83 (5.1) | 3,200 (2,180–4,280) | 163 (10.0) | NA | NA | NA |
| Co.N.ER, *n* = 641, 2004–2005 (Italy) | 549 (85.6) | 30 (4.7) | 62 (9.7) | 441 (68.9) | 199 (31.1) | 39.0 (36.0–41.0) | 29 (4.5) | 3,340 (2,420–4,230) | 63 (9.9) | 95.0 (86.6–111.1) | 0.69 (−1.29 to 2.92) | 102 (35.5) |
| DNBC, *n* = 71,710, 1996–2002 (Denmark) | 59,030 (82.3) | NA | 12,680 (17.7) | 49,534 (70.5) | 20,756 (29.5) | 40.1 (35.9–42.4) | 3,168 (4.4) | 3,600 (2,420–4,640) | 7,124 (10.0) | 85.0 (75.1–89.5) | 0.01 (−1.95 to 2.07) | 5,644 (15.5) |
| EDEN, *n* = 1,880, 2003–2005 (France) | 1,376 (73.2) | 148 (7.9) | 356 (18.9) | 999 (59.5) | 679 (40.5) | 39.0 (35.0–41.0) | 106 (5.6) | 3,300 (2,158–4,200) | 187 (10.0) | 67.6 (65.0–72.4) | −0.01 (−1.52 to 2.02) | 145 (12.9) |
| FCOU, *n* = 4,003, 1993–1996 (Ukraine) | 3,647 (91.1) | NA | 356 (8.9) | 461 (16.2) | 2,382 (83.8) | NA | NA | 3,400 (2,100–4,300) | 393 (10.2) | 84.0 (75.0–93.0) | −0.02 (−2.02 to 2.06) | 119 (12.7) |
| GASPII, *n* = 680, 2003–2004 (Italy) | 599 (88.1) | 23 (3.4) | 58 (8.5) | 510 (75.2) | 168 (24.8) | 40.0 (36.0–42.0) | 28 (4.1) | 3,350 (2,401–4,320) | 67 (9.9) | 104.0 (98.0–113.0) | 0.70 (−1.37 to 2.66) | 172 (37.1) |
| GENERATION R, *n* = 7,934, 2002–2006 (The Netherlands) | 6,190 (78.0) | 461 (5.8) | 1,283 (16.2) | 2,833 (56.5) | 2,183 (43.5) | 40.1 (35.4–42.3) | 474 (6.0) | 3,420 (2,190–4,480) | 788 (10.0) | 115.3 (69.4–119.4) | 0.35 (−1.52 to 2.67) | 1,578 (27.1) |
| GENERATION XXI, *n* = 7,541, 2005–2006 (Portugal) | 5,766 (76.5) | 540 (7.2) | 1,235 (16.4) | NA | NA | 39.0 (35.0–41.0) | 557 (7.4) | 3,200 (2,130–4,095) | 747 (10.0) | 85.0 (70.2–95.0) | 0.63 (−1.38 to 3.23) | 1,991 (37.9) |
| GENESIS, *n* = 2,261, 2003–2004 (Greece) | 1,842 (81.5) | 30 (1.3) | 389 (17.2) | NA | NA | 40.0 (34.0–40.0) | 224 (10.0) | 3,250 (2,100–4,200) | 213 (10.0) | 61.9 (60.1–71.9) | 0.93 (−1.43 to 4.11) | 39 (43.3) |
| GINIplus, *n* = 2,086, 1995–1998 (Germany) | 1,903 (91.2) | NA | 193 (8.8) | NA | NA | NA | NA | NA | NA | 62.9 (60.2–74.4) | 0.01 (−1.77 to 1.93) | 215 (10.3) |
| HUMIS, *n* = 986, 2002–2009 (Norway) | 932 (94.5) | NA | 54 (5.5) | NA | NA | 40.1 (33.2–42.9) | 86 (8.7) | 3,580 (1,822–4,703) | 98 (10.0) | 84.0 (60.0–92.0) | 0.02 (−2.03 to 2.14) | 58 (17.5) |
| INMA, *n* = 2,406, 1997–2008 (Spain) | 1,988 (82.6) | NA | 418 (17.4) | 1,395 (58.0) | 1,009 (42.0) | 39.9 (36.0–42.0) | 98 (4.1) | 3,250 (2,300–4,200) | 238 (10.0) | 83.6 (75.1–94.5) | 0.55 (−1.37 to 3.31) | 489 (37.7) |
| KOALA, *n* = 2,800, 2000–2002 (the Netherlands) | 2,594 (92.6) | NA | 206 (7.4) | NA | NA | 40.0 (36.0–42.0) | 89 (3.2) | 3,500 (2,478–4,510) | 277 (10.0) | 106.2 (61.5–119.3) | −0.17 (−2.16 to 1.77) | 199 (11.4) |

(*Continued*)

**Table 1.** (Continued)

| Cohort name, number of participants, birth years (country) | Maternal smoking | | | Paternal smoking | | Birth outcomes | | | | Childhood BMI | | |
|---|---|---|---|---|---|---|---|---|---|---|---|---|
| | No | First trimester only | Continued | No | Yes | Gestational age at birth (weeks) | Preterm birth | Birth weight (g) | Small size for gestational age at birth | Age (months) | BMI (SDS) | Overweight |
| LISAplus, n = 1,965, 1997–1999 (Germany) | 1,697 (86.4) | 87 (4.4) | 181 (9.2) | 1,557 (82.0) | 342 (18.0) | NA | NA | NA | NA | 62.7 (60.2–74.0) | −0.09 (−1.92 to 1.88) | 201 (10.2) |
| LUKAS, n = 441, 2002–2005 (Finland) | 371 (84.1) | 35 (7.9) | 35 (7.9) | NA | NA | NA | NA | 3,630 (2,790–4,689) | 44 (10.0) | 73.2 (68.6–76.0) | 0.52 (−1.08 to 3.33) | 114 (31.4) |
| MoBa, n = 80,116, 1999–2009 (Norway) | 72,466 (90.5) | NA | 7,650 (9.5) | 63,071 (79.2) | 16,523 (20.8) | 40.1 (36.1–42.4) | 3,312 (4.1) | 3,620 (2,521–4,640) | 7,967 (10.0) | 85.9 (61.0–100.9) | 0.15 (−2.05 to 2.30) | 6,002 (19.5) |
| NINFEA, n = 2,259, 2005–2010 (Italy)[a] | 2,085 (92.3) | 29 (1.3) | 145 (6.4) | NA | NA | 39.7 (35.9–41.9) | 91 (4.0) | 3,240 (2,271–4,189) | 220 (10.0) | 86.1 (84.8–93.1) | −0.02 (−2.16 to 2.43) | 95 (21.5) |
| PÉLAGIE, n = 1,353, 2002–2005 (France) | 1,022 (75.2) | 172 (12.7) | 159 (11.8) | 597 (61.8) | 369 (38.2) | 40.0 (36.0–41.0) | 44 (3.3) | 3,400 (2,460–4,315) | 135 (10.0) | NA | NA | NA |
| Piccolipiù, n = 3,292, 2011–2015 (Italy) | 2,572 (78.1) | 374 (11.4) | 346 (10.5) | 1,496 (71.4) | 598 (28.6) | 39.0 (36.0–41.0) | 93 (2.9) | 3,340 (2,470–4,229) | 323 (10.0) | NA | NA | NA |
| PRIDE Study, n = 1,616, 2011–2015 (the Netherlands) | 1,519 (94.0) | 39 (2.4) | 58 (3.6) | NA | NA | 39.0 (35.6–41.0) | 77 (4.9) | 3,484 (2,280–4,500) | 154 (9.9) | NA | NA | NA |
| Project Viva, n = 2,001, 1999–2002 (USA) | 1,784 (89.2) | 124 (6.2) | 93 (4.6) | NA | NA | 39.7 (34.7–41.9) | 142 (7.1) | 3,487 (2,155–4,536) | 199 (10.0) | 92.2 (82.5–116.5) | 0.42 (−1.38 to 3.04) | 315 (30.6) |
| REPRO_PL, n = 1,434, 2007–2011 (Poland) | 1,215 (84.7) | 83 (5.8) | 136 (9.5) | 866 (63.0) | 509 (37.0) | 39.0 (36.0–41.0) | 64 (4.5) | 3,350 (2,376–4,290) | 142 (10.0) | 88.0 (84.3–94.0) | 0.64 (−1.55 to 3.64) | 19 (38.8) |
| RHEA, n = 651, 2007–2008 (Greece) | 544 (83.6) | NA | 107 (16.4) | 287 (48.6) | 303 (51.4) | 38.0 (35.0–40.0) | 73 (11.3) | 3,190 (2,312–4,059) | 63 (9.9) | NA | NA | NA |
| SCOPE BASELINE, n = 1,216, 2009–2011 (Ireland) | 1,078 (88.7) | NA | 138 (11.3) | 739 (78.5) | 203 (21.5) | 40.3 (35.2–41.7) | 60 (4.9) | 3,460 (2,353–4,485) | 121 (10.0) | NA | NA | NA |
| SWS, n = 2,716, 1998–2007 (UK) | 2,316 (85.3) | NA | 400 (14.7) | NA | NA | 40.1 (35.1–42.1) | 154 (5.7) | 3,450 (2,330–4,475) | 268 (10.0) | 80.3 (74.7–87.2) | 0.21 (−1.51 to 2.47) | 368 (22.0) |
| **Total group** | 196,168 (85.6) | 2,247 (1.0) | 30,743 (13.4) | 134,939 (72.5) | 51,084 (27.5) | 40.0 (35.7–42.3) | 10,299 (4.7) | 3,530 (2,390–4,580) | 22,312 (10.0) | 85.2 (61.0–117.7) | 0.13 (−1.86 to 2.43) | 21,345 (20.0) |

Values are expressed as number of participants (valid %) or medians (95% range). "First trimester only" refers to mothers who smoked during first trimester only. Childhood overweight also includes obesity and includes information at child age ≥5 to <10 years. Preterm birth is defined as birth before the gestational age of 37 weeks. Small size for gestational age is defined as the lowest 10% of sex- and gestational age–adjusted birth weight SDS per cohort.

[a] Subset of participants with follow-up completed at 4 years of child's age by the time of data transfer (March 2015).

Abbreviations: BMI, body mass index; NA, not available (not collected or not provided) or not applicable (gestational age at birth [FCOU, GINIplus, LISAplus, LUKAS] and birth weight [GINIplus, LISAplus] due to study samples restricted to specific ranges of gestational age and weight at birth); SDS, standard deviation score

**Table 2. Maternal smoking with risks of birth complications and childhood overweight.**

| Maternal smoking | Preterm birth | Small size for gestational age at birth | Childhood overweight |
|---|---|---|---|
| | Odds ratio (95% confidence interval) | Odds ratio (95% confidence interval) | Odds ratio (95% confidence interval) |
| **No maternal smoking** | Reference | Reference | Reference |
| | $n_{cases/total}$ = 8,586/188,357 | $n_{cases/total}$ = 16,879/190,873 | $n_{cases/total}$ = 17,530/92,434 |
| **Only first trimester smoking** | 1.03 (0.85–1.25) | 0.99 (0.85–1.15) | 1.17 (1.02–1.35)* |
| | $n_{cases/total}$ = 120/2,116 | $n_{cases/total}$ = 200/2,144 | $n_{cases/total}$ = 329/1,084 |
| **First trimester dosage** | | | |
| ≤4 cigarettes/day | 0.99 (0.70–1.39) | 0.96 (0.75–1.22) | 1.02 (0.78–1.33) |
| | $n_{cases/total}$ = 36/828 | $n_{cases/total}$ = 77/826 | $n_{cases/total}$ = 78/340 |
| 5–9 cigarettes/day | 1.00 (0.58–1.72) | 0.90 (0.59–1.36) | 1.37 (0.92–2.06) |
| | $n_{cases/total}$ = 14/288 | $n_{cases/total}$ = 25/288 | $n_{cases/total}$ = 35/136 |
| ≥10 cigarettes/day | 0.81 (0.45–1.46) | 0.88 (0.57–1.35) | 1.31 (0.89–1.93) |
| | $n_{cases/total}$ = 12/273 | $n_{cases/total}$ = 23/271 | $n_{cases/total}$ = 40/152 |
| **Continued smoking** | 1.08 (1.02–1.15)* | 2.15 (2.07–2.23)** | 1.42 (1.35–1.48)** |
| | $n_{cases/total}$ = 1,593/29,951 | $n_{cases/total}$ = 5,233/30,125 | $n_{cases/total}$ = 3,486/13,083 |
| **Continued smoking dosage** | | | |
| ≤4 cigarettes/day | 1.01 (0.89–1.14) | 1.57 (1.45–1.70)** | 1.30 (1.18–1.42)** |
| | $n_{cases/total}$ = 288/5,866 | $n_{cases/total}$ = 836/6,034 | $n_{cases/total}$ = 688/2,792 |
| 5–9 cigarettes/day | 1.07 (0.95–1.19) | 2.40 (2.25–2.56)** | 1.42 (1.30–1.55)** |
| | $n_{cases/total}$ = 367/7,115 | $n_{cases/total}$ = 1,341/7,162 | $n_{cases/total}$ = 813/3,284 |
| ≥10 cigarettes/day | 1.11 (1.01–1.22)* | 2.93 (2.76–3.10)** | 1.55 (1.43–1.67)** |
| | $n_{cases/total}$ = 524/9,771 | $n_{cases/total}$ = 2,001/9,743 | $n_{cases/total}$ = 1,137/4,139 |

Values are odds ratios (95% confidence intervals) from multilevel binary logistic mixed-effects models that reflect the risk of preterm birth, small size for gestational age, and childhood overweight per smoking group compared with the reference group (no maternal smoking).

Number of cigarettes used as continued smoking dosage was based on third trimester information. Preterm birth is defined as birth before the gestational age of 37 weeks. Small size for gestational age is defined as the lowest 10% of sex- and gestational age–adjusted birth weight standard deviation score per cohort. Childhood overweight is overweight and obesity together according to the World Health Organization criteria. Models are adjusted for maternal age, educational level, parity, prepregnancy or early-pregnancy body mass index, alcohol consumption during pregnancy, and paternal smoking.

*$P$ value < 0.05.

**$P$ value < 0.001.

## Parental smoking during pregnancy and the risks of preterm birth, SGA, and childhood overweight

Among mothers who did not smoke during pregnancy, paternal smoking tended to be associated with higher risks of preterm birth (OR 1.06 [1.00–1.12], $P$ value = 0.05), SGA (OR 1.04 [1.00–1.09], $P$ value = 0.05), and childhood overweight (OR 1.21 [1.16–1.27], $P$ value < 0.001) (**Table 4**). Among mothers who smoked during first trimester only, paternal smoking was not associated with preterm birth or SGA but was associated with a higher risk of childhood overweight (OR 1.36 [1.02–1.80], $P$ value = 0.036). Among mothers who continued smoking during pregnancy, paternal smoking further increased the risks of SGA and childhood overweight (both Z-score $P$ value for differences in effect sizes between categories <0.0001) but not the risk of preterm birth. Children whose mothers continued smoking during pregnancy and whose fathers also smoked had the highest risks of being born preterm (OR 1.10 [1.02–1.19], $P$ value = 0.016) and SGA (OR 2.37 [2.26–2.49], $P$ value < 0.001) and of childhood overweight (OR 1.76 [1.65–1.87], $P$ value < 0.001). Similar results were observed for the combined maternal and paternal smoking with the continuous outcomes (S6 Table).

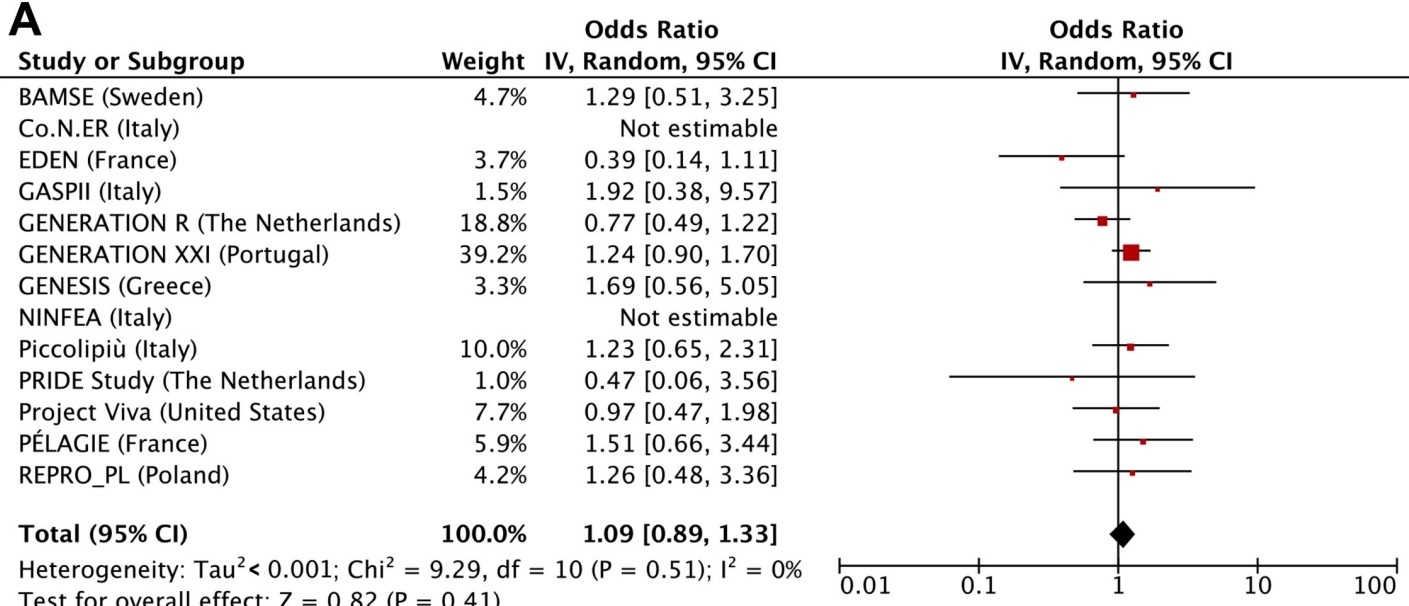

**Fig 2. Maternal smoking with risks of preterm birth assessed by 2-stage random-effects models.** (A) First trimester smoking versus nonsmoking, (B) continued smoking versus nonsmoking. Values are odds ratios (95% CIs) per cohort and pooled from binary logistic regression models that reflect the risk of preterm birth per smoking pattern (first-trimester-only smoking or continued smoking) compared to that of nonsmoking. Models are adjusted for maternal age, educational level, parity,

prepregnancy or early-pregnancy body mass index, alcohol consumption during pregnancy, and paternal smoking. The cohorts for which no estimate was provided had no data available for that particular analysis. The heterogeneity between the estimates of each cohort was 0% (95% CI 0%–57%) and 4% (95% CI 0%–47%) for first-trimester-only smoking and continued smoking, respectively. CI, confidence interval, IV, instrumental variable.

## Discussion

In this study, maternal continued smoking during pregnancy was associated, in a dose-response manner, to higher risks of preterm birth, being SGA at birth, and childhood over-weight. Maternal smoking during the first trimester of pregnancy only was not associated with risks of preterm birth and SGA but was associated with a higher risk of childhood overweight. Reducing the number of cigarettes during pregnancy without quitting may be beneficial for the risk of SGA but seems not to influence the risks of preterm birth and childhood over-weight. Paternal smoking seems to be associated, independently of maternal smoking, with the risks of childhood overweight.

Maternal smoking is a major public health concern [1]. The associations of maternal con-tinued smoking during pregnancy and increased risks of preterm birth and SGA are well established [7,10,18]. Also, several studies have suggested associations of fetal smoke exposure with childhood overweight and obesity [11,22]. In line with these previous studies, we observed that children whose mothers continued smoking during pregnancy have higher risks of preterm birth, being SGA at birth, and overweight in childhood. The risks of preterm birth were somewhat weaker than reported previously [7,9,18], potentially because no information was available about induced or spontaneous preterm birth.

Results from previous studies focused on the associations of maternal early smoking cessa-tion and of reducing the number of cigarettes during pregnancy with child health outcomes are inconsistent [8,17,19,21,22]. Results from prospective studies in the Netherlands and Aus-tralia previously suggested that quitting smoking after the first trimester was not associated with risks of adverse birth outcomes [18,19]. A large US study with more than 21,000 first tri-mester smokers reported that smoking of any duration during pregnancy was associated with an increased risk of fetal growth restriction with decreasing risk the earlier that cessation occurred [17]. Similarly, a recent study from the UK Millennium Cohort Study suggested that two-thirds of the total adverse smoking impact on birth weight occurs in the second trimester and that cutting smoking intensity by the third month in pregnancy leads to infants of the same weight as those infants born to persistent light smokers [37]. A recent study investigating associations of parental smoking with fetal growth using additional methods of mendelian ran-domization and parental negative control showed consistent linear dose-dependent associa-tions of maternal smoking with fetal growth from early second trimester onward [38]. These studies suggest that smoking cessation programs should focus on the benefit of quitting as early in pregnancy as possible. A previous analysis using data from the Nurses' Health Study showed that first-trimester-only maternal smoking was not, or was only to a limited extent, associated with obesity in later life [20]. However, in the same cohort, first-trimester-only maternal smoking was associated with type 2 diabetes in the offspring [39]. In the current study, maternal first-trimester-only smoking was not associated with the risks of preterm birth or SGA but was associated with an increased risk of childhood overweight. A biological expla-nation might be that maternal first-trimester-only smoking already leads to specific adapta-tions, which might have lifelong consequences for body composition and metabolic health in later life, but the fetal smoke exposure is not long enough to affect birth outcomes. Reducing the number of cigarettes from first to third trimester lowered the risks of SGA, but risks were still elevated compared with those in infants born to nonsmoking mothers. This association was not observed for preterm birth and childhood overweight. Thus, our findings suggest that

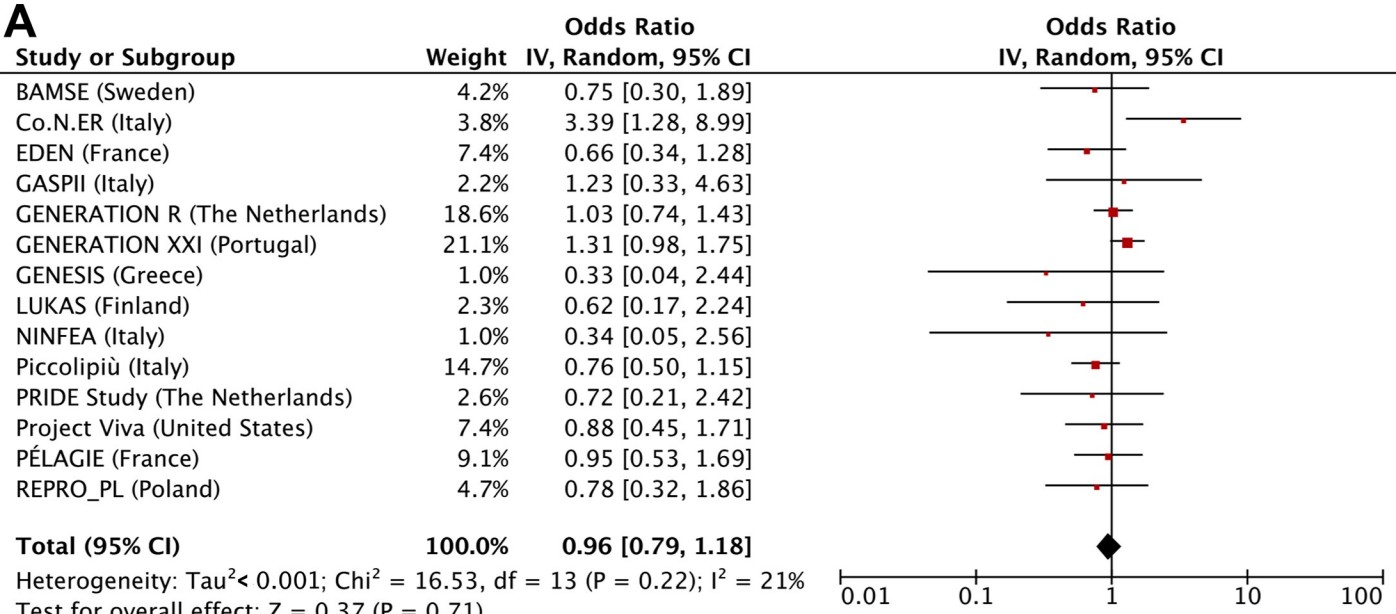

**Fig 3. Maternal smoking with risks of small size for gestational age assessed by two-stage random-effects models.** (A) First trimester smoking versus nonsmoking, (B) continued smoking versus nonsmoking. Values are odds ratios (95% CIs) per cohort and pooled from binary logistic regression models that reflect the risk of small size for gestational age per smoking pattern (first-trimester-only smoking or continued smoking) compared to that of nonsmoking. Models are adjusted for maternal

age, educational level, parity, prepregnancy or early-pregnancy body mass index, alcohol consumption during pregnancy, and paternal smoking. The cohorts for which no estimate was provided had no data available for that particular analysis. The heterogeneity between the estimates of each cohort was 21% (95% CI 0%–65%) and 75% (95% CI 56%–86%) for first-trimester-only smoking and continued smoking, respectively. CI, confidence interval, IV, instrumental variable.

quitting smoking in the first trimester of pregnancy might optimize birth outcomes but might not reduce the risk of adverse metabolic effects in the offspring to the level of nonsmoking. Also, reducing the number of cigarettes from first trimester onward may reduce risks of fetal growth restriction.

The role of paternal smoking during pregnancy on child health outcomes remains unclear [23,40,41]. Paternal smoking has been associated with reduced semen quality and fertility and higher risks of spontaneous abortion, birth defects, and, in the long-term, attention-deficit/hyperactivity disorder and several cancers [42–45]. A recent meta-analysis showed that paternal smoking was associated with increased risks of preterm birth and SGA [44]. In a previous Dutch study, paternal smoking during pregnancy among nonsmoking mothers was associated with higher childhood BMI [12]. A small study from the US using self-reported smoking and serum cotinine measurements found a higher BMI at 2 and 3 years of age in children whose mothers were exposed to passive smoking during pregnancy [40]. In the current study, paternal smoking among nonsmoking mothers was associated with a higher risk of childhood overweight and tended to be associated with higher risks of preterm birth and SGA. This suggests that paternal smoking may be, independently of maternal smoking, associated with childhood overweight. However, we cannot exclude the possibility of residual confounding by factors not or insufficiently measured in the studies. Previous studies used comparisons of maternal and paternal smoking associations to explore potential mechanisms [12,46]. In the current study, if only one parent smoked, the risks of SGA were much higher among maternal smokers than among paternal smokers, whereas the risks of preterm birth for maternal and paternal smoking were similar. The similar associations of maternal and paternal smoking and preterm birth may suggest that the underlying mechanisms include shared family-based characteristics, such as environmental exposures and lifestyle. The stronger associations of maternal smoking, compared with paternal smoking, with SGA may suggest that these associations are mainly explained by intrauterine mechanisms. Since paternal smoking among nonsmoking mothers was not associated with SGA, the risk increase when both parents smoked may represent an additional mechanistic pathway through shared family-based characteristics. The risk of overweight was slightly higher among children whose mothers smoked than whose fathers smoked. However, the risks increased significantly if both parents smoked. These findings suggest that, although intrauterine programming mechanisms might play a role, shared family-based lifestyle and genetic characteristics are potential underlying mechanisms. Whether these findings also reflect transgenerational epigenetic inheritance through the gametes needs to be further studied.

Various components of tobacco smoke might be involved in the mechanistic pathway toward adverse birth outcomes and childhood overweight. Both nicotine and carbon monoxide are reported to reduce placental blood flow [47]. Nicotine stimulates acetylcholine receptors, which release a multitude of vasoactive catecholamines and peptides, which in turn reduce blood flow through vasoconstriction [47]. Carbon monoxide competes with oxygen for binding sites on the transport protein hemoglobin, causing hypoxia [48]. Chronic hypoxia interferes with the maternal circulatory adjustments to pregnancy which can be another cause of reduced placental blood flow [49]. Uterine blood flow is essential for uterine, placental, and fetal growth. Several mechanisms for nicotine-induced alterations in overweight risks have been proposed, including stimulation of the fetal hypothalamic-pituitary axis [50]. It has been

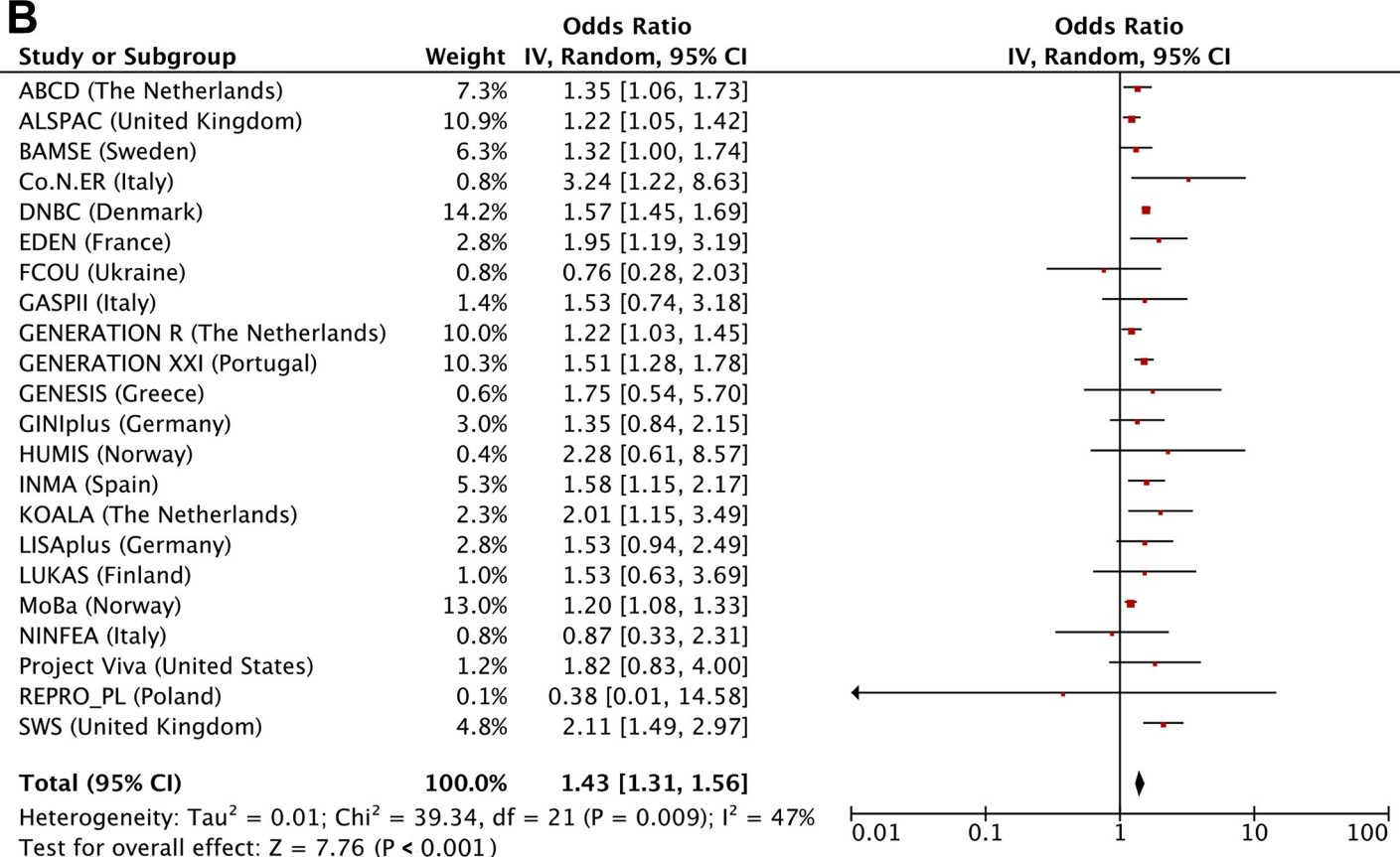

**Fig 4. Maternal smoking with risks of childhood overweight assessed by two-stage random-effects models.** (A) First trimester smoking versus nonsmoking, (B) continued smoking versus nonsmoking. Values are odds ratios (95% CIs) per cohort and pooled from binary logistic regression models that reflect the risk of childhood overweight per smoking pattern (first-trimester-only smoking or continued smoking) compared to that of nonsmoking. Models are adjusted for maternal age, educational level, parity, prepregnancy or early-pregnancy body mass index, alcohol consumption during pregnancy, and paternal smoking. The cohorts for which no estimate was provided had no data available for that particular analysis. The heterogeneity between the estimates of each cohort was 0% (95% CI 0%–60%) and 47% (95% CI 1%–72%) for first-trimester-only smoking and continued smoking, respectively. CI, confidence interval, IV, instrumental variable.

**Table 3. Change in maternal smoking habits during pregnancy and risks of birth complications and childhood overweight.**

| Maternal smoking | Preterm birth | Small size for gestational age at birth | Childhood overweight |
|---|---|---|---|
| | Odds ratio (95% confidence interval) | Odds ratio (95% confidence interval) | Odds ratio (95% confidence interval) |
| **No maternal smoking in first trimester** | | | |
| Third trimester no smoking | Reference | Reference | Reference |
| | $n_{\text{cases/total}}$ = 4,527/100,634 | $n_{\text{cases/total}}$ = 8,698/103,740 | $n_{\text{cases/total}}$ = 11,177/59,070 |
| Third trimester ≤4 cigarettes/day | 0.73 (0.40–1.34) | 1.20 (0.81–1.78) | 1.31 (0.90–1.92) |
| | $n_{\text{cases/total}}$ = 11/278 | $n_{\text{cases/total}}$ = 28/274 | $n_{\text{cases/total}}$ = 41/147 |
| Third trimester 5–9 cigarettes/day | 1.07 (0.48–2.48) | 2.02 (1.18–3.46)* | 1.27 (0.64–2.37) |
| | $n_{\text{cases/total}}$ = 6/104 | $n_{\text{cases/total}}$ = 16/103 | $n_{\text{cases/total}}$ = 13/51 |
| Third trimester ≥10 cigarettes/day | 1.51 (0.65–3.49) | 1.74 (0.91–3.32) | 1.60 (0.72–3.55) |
| | $n_{\text{cases/total}}$ = 6/80 | $n_{\text{cases/total}}$ = 11/79 | $n_{\text{cases/total}}$ = 10/31 |
| **Maternal smoking in first trimester ≤4 cigarettes/ day** | | | |
| Third trimester quit | 0.96 (0.69–1.35) | 1.04 (0.82–1.31) | 1.20 (0.94–1.53) |
| | $n_{\text{cases/total}}$ = 38/862 | $n_{\text{cases/total}}$ = 84/859 | $n_{\text{cases/total}}$ = 98/388 |
| Third trimester ≤4 cigarettes/day | 1.05 (0.86–1.27) | 1.54 (1.37–1.74)** | 1.32 (1.14–1.52)** |
| | $n_{\text{cases/total}}$ = 114/2,261 | $n_{\text{cases/total}}$ = 328/2,457 | $n_{\text{cases/total}}$ = 289/1,169 |
| Third trimester 5–9 cigarettes/day | 1.15 (0.85–1.55) | 2.43 (2.05–2.89)** | 1.81 (1.45–2.25)** |
| | $n_{\text{cases/total}}$ = 47/885 | $n_{\text{cases/total}}$ = 170/880 | $n_{\text{cases/total}}$ = 121/440 |
| Third trimester ≥10 cigarettes/day | 1.37 (0.76–2.47) | 2.47 (1.71–3.58)** | 1.31 (0.79–2.19) |
| | $n_{\text{cases/total}}$ = 12/186 | $n_{\text{cases/total}}$ = 36/185 | $n_{\text{cases/total}}$ = 21/86 |
| **Maternal smoking in first trimester 5–9 cigarettes/ day** | | | |
| Third trimester quit | 1.04 (0.62–1.73) | 0.95 (0.64–1.42) | 1.32 (0.91–1.92) |
| | $n_{\text{cases/total}}$ = 16/304 | $n_{\text{cases/total}}$ = 27/304 | $n_{\text{cases/total}}$ = 41/165 |
| Third trimester ≤4 cigarettes/day | 0.86 (0.58–1.28) | 1.89 (1.52–2.34)** | 1.53 (1.17–2.00)* |
| | $n_{\text{cases/total}}$ = 27/657 | $n_{\text{cases/total}}$ = 102/654 | $n_{\text{cases/total}}$ = 80/307 |
| Third trimester 5–9 cigarettes/day | 1.00 (0.85–1.18) | 2.21 (2.02–2.42)** | 1.43 (1.26–1.61)** |
| | $n_{\text{cases/total}}$ = 163/3,551 | $n_{\text{cases/total}}$ = 630/3,617 | $n_{\text{cases/total}}$ = 403/1,704 |
| Third trimester ≥10 cigarettes/day | 0.99 (0.76–1.30) | 2.70 (2.35–3.10)** | 1.40 (1.15–1.69)* |
| | $n_{\text{cases/total}}$ = 59/1,330 | $n_{\text{cases/total}}$ = 265/1,319 | $n_{\text{cases/total}}$ = 149/632 |
| **Maternal smoking in first trimester ≥10 cigarettes/ day** | | | |
| Third trimester quit | 0.82 (0.46–1.43) | 1.06 (0.71–1.57) | 1.34 (0.96–1.88) |
| | $n_{\text{cases/total}}$ = 13/285 | $n_{\text{cases/total}}$ = 28/283 | $n_{\text{cases/total}}$ = 52/194 |
| Third trimester ≤4 cigarettes/day | 1.26 (0.82–1.95) | 1.93 (1.46–2.57)** | 1.14 (0.81–1.61) |
| | $n_{\text{cases/total}}$ = 22/358 | $n_{\text{cases/total}}$ = 59/354 | $n_{\text{cases/total}}$ = 48/192 |
| Third trimester 5–9 cigarettes/day | 1.26 (0.97–1.63) | 2.79 (2.39–3.25)** | 1.46 (1.18–1.80)** |
| | $n_{\text{cases/total}}$ = 62/1,078 | $n_{\text{cases/total}}$ = 224/1,072 | $n_{\text{cases/total}}$ = 128/503 |

(*Continued*)

**Table 3.** (Continued)

| Maternal smoking | Preterm birth | Small size for gestational age at birth | Childhood overweight |
|---|---|---|---|
| | Odds ratio (95% confidence interval) | Odds ratio (95% confidence interval) | Odds ratio (95% confidence interval) |
| Third trimester ≥10 cigarettes/day | 1.16 (1.04–1.31)* | 2.95 (2.75–3.15)** | 1.67 (1.53–1.83)** |
| | $n_{cases/total}$ = 364/6,949 | $n_{cases/total}$ = 1,434/6,940 | $n_{cases/total}$ = 849/2,976 |

Values are odds ratios (95% confidence intervals) from multilevel binary logistic mixed-effects models that reflect the risk of preterm birth, small size for gestational age, and childhood overweight per change in smoking group compared with that of the reference group (nonsmoking in first and third trimester). Preterm birth is defined as birth before the gestational age of 37 weeks. Small size for gestational age is defined as the lowest 10% of sex- and gestational age–adjusted birth weight standard deviation score per cohort. Childhood overweight is overweight and obesity together according to the World Health Organization criteria. Models are adjusted for maternal age, educational level, parity, prepregnancy body mass index, alcohol consumption during pregnancy, and paternal smoking.

*P value < 0.05.

**P value < 0.001.

suggested that cadmium, present in tobacco smoke, modulates oxytocin receptor function, proposing a role in the pathophysiology of preterm birth [48]. Recent studies have found an association between maternal smoking during pregnancy and birth weight with a mediating role of DNA methylation [51–53]. Further research is needed to assess such possible mechanisms. During the last few years, e-cigarettes have been widely used as substitutes for smoking. Evidence from recently started cohorts is needed to clarify whether e-cigarettes are any safer during pregnancy.

We performed an individual participant data meta-analysis of prospective cohort studies to investigate the associations of parental smoking during pregnancy with preterm birth, SGA, and childhood overweight. We included data from cohort studies in Europe and North America, so our findings are mainly applicable to populations in developed countries. Inclusion of data from other regions could have led to differences in prevalence of maternal and paternal smoking, birth complications, childhood overweight, and ethnic and sociodemographic characteristics, complicating or limiting the possibility of doing a meta-analysis. Among study limitations, our outcomes might not be generalizable to populations from low-income and middle-income countries, which need to be further studied. The large sample size enabled us to investigate the effects of changing smoking habits and paternal smoking. However, our study might have been underpowered to detect associations in the analyses looking at maternal-only first trimester smoking and the change in smoking habits from first to third trimester, due to small sample sizes. Since we used original, individual participant data, we did not formally assess the quality of the individual studies included. We are aware that our study cannot overcome potential limitations of individual studies in terms of their design and conduct, differences in the definitions of exposure and outcome data, and variation in missing data. Parental smoking information during pregnancy was self-reported. For active smoking, correlations between cotinine measurements and self-reported smoking habits are high [54]. We have no information on the specific question asked or the timing in which it was asked, which might have differed across cohorts and influenced our results. It has been suggested that using maternal nonsmokers as a reference group without considering the impact of passive smoke exposure may contribute to an underestimation of the estimated effects [40]. To limit this misclassification, all analyses on maternal smoking were adjusted for paternal smoking.

**Table 4. Associations of maternal and paternal smoking with risks of birth complications and childhood overweight.**

| Maternal and paternal smoking | Preterm birth | Small size for gestational age at birth | Childhood overweight |
|---|---|---|---|
| | Odds ratio (95% confidence interval) | Odds ratio (95% confidence interval) | Odds ratio (95% confidence interval) |
| **Maternal nonsmoking** | | | |
| Paternal nonsmoking | Reference | Reference | Reference |
| | $n_{cases/total} = 5,232/123,666$ | $n_{cases/total} = 10,746/123,328$ | $n_{cases/total} = 10,298/59,395$ |
| Paternal smoking | 1.06 (1.00–1.12) | 1.04 (1.00–1.09) | 1.21 (1.16–1.27)** |
| | $n_{cases/total} = 1,505/31,890$ | $n_{cases/total} = 3,030/33,691$ | $n_{cases/total} = 3,199/15,474$ |
| **Maternal first trimester smoking** | | | |
| Paternal nonsmoking | 0.64 (0.36–1.15) | 0.78 (0.53–1.13) | 1.36 (0.98–1.87) |
| | $n_{cases/total} = 12/412$ | $n_{cases/total} = 30/412$ | $n_{cases/total} = 54/233$ |
| Paternal smoking | 1.03 (0.70–1.51) | 1.05 (0.80–1.39) | 1.36 (1.02–1.80)* |
| | $n_{cases/total} = 29/626$ | $n_{cases/total} = 59/625$ | $n_{cases/total} = 70/305$ |
| **Maternal continued smoking** | | | |
| Paternal nonsmoking | 1.04 (0.93–1.15) | 2.06 (1.94–2.20)** | 1.33 (1.23–1.44)** |
| | $n_{cases/total} = 405/8,768$ | $n_{cases/total} = 1,366/8,723$ | $n_{cases/total} = 877/3,872$ |
| Paternal smoking | 1.10 (1.02–1.19)* | 2.37 (2.26–2.49)** | 1.76 (1.65–1.87)** |
| | $n_{cases/total} = 810/15,806$ | $n_{cases/total} = 2,896/15,967$ | $n_{cases/total} = 1,785/6,661$ |

Values are odds ratios (95% confidence intervals) from multilevel binary logistic mixed-effects models that reflect the risk of preterm birth, small size for gestational age, and childhood overweight per smoking group compared with the reference group (no parental smoking).

Preterm birth is defined as birth before the gestational age of 37 weeks. Small size for gestational age is defined as the lowest 10% of sex- and gestational age–adjusted birth weight standard deviation score per cohort. Childhood overweight is overweight and obesity together according to the World Health Organization criteria. Models are adjusted for maternal age, maternal body mass index, paternal body mass index, maternal education, parity, and maternal alcohol consumption during pregnancy.

*$P$ value $< 0.05$.

**$P$ value $< 0.001$.

Although smoking in the preconception period has been reported not to be associated with fetal growth restriction, studies considering its effect on childhood overweight are lacking [17]. In the current study, information on smoking in the preconception period was missing. Further research is needed to assess the associations of smoking in the preconception period with offspring outcomes. It has been suggested that exposure to smoking during childhood amplifies the association between prenatal smoke exposure and childhood BMI outcomes [55]. Many women resume smoking shortly after birth. Six weeks after birth, approximately 25% of women resumed smoking, and 1 year after birth these numbers are up to 80% [56]. In our study, information on exposure to smoking during childhood was not available for most cohorts. Further research is needed to assess whether childhood BMI outcomes are additionally influenced by exposure to smoking during childhood. Overall, we observed low to moderate heterogeneity in the 2-stage random-effects models, which might be due to the inclusion of cohorts that were mostly high-income and of European descent. However, we observed high heterogeneity between the cohorts for the associations of maternal continued smoking with SGA. This might be in part explained by differences in pattern and dosage of maternal and paternal smoking between cohorts. When we restricted the 2-stage continued smoking models to the cohorts that also had information on first-trimester-only smoking, we observed a substantially lower heterogeneity between estimates. Missing values of covariates were used as an additional group. This approach has been commonly used in large meta-analyses of individual participant data because of the constraints in applying more advanced imputation strategies. Although we cannot disregard the possibility of bias, we consider it unlikely considering the relatively small percentage of missing data [57]. We observed similar results when we

conducted a complete case analysis (S7 Table). Also, similar associations were observed when adjusting for maternal age and BMI as categorical or continuous covariates (S7 and S8 Tables). Although we adjusted for multiple lifestyle-related factors, we cannot exclude residual confounding by other environmental lifestyle-related factors. From the current observational data, no conclusions can be drawn on the causality of the observed associations.

Our results suggest that as compared to mothers who continued smoking throughout pregnancy, mothers who quit smoking during the first trimester have a reduced risk of birth complications. Reducing the number of cigarettes without quitting during pregnancy is still associated with an increased risk of birth complications. The observed risk estimates were small to moderate but are important from a public health perspective, since smoking is a common adverse exposure and preterm birth and SGA are among the most frequent birth complications. Also, preterm birth, SGA, and childhood obesity are related with adverse health consequences later in life. Our findings suggest that it is of great importance to invest in prevention of smoking in women of reproductive age before or at the start of pregnancy. Pregnant women should still be motivated to reduce smoking, even later in pregnancy. The current guidelines focus only on quitting smoking and not reducing, which can be discouraging for women who find it difficult to quit smoking. These women should be provided with sufficient information about the risks of continued smoking but also about the benefits of reducing their number of cigarettes. Future research should investigate whether quitting smoking in the first trimester or reducing the number of cigarettes during pregnancy is also beneficial for other adverse birth and offspring outcomes. Although we cannot exclude a role of residual confounding and shared family-based characteristics in the associations of paternal smoking with childhood overweight, we recommend that fathers are more closely involved in preconception and pregnancy consultations focused on smoking reduction.

Our results suggest that maternal smoking during the first trimester only is not associated with the risks of SGA and preterm birth but is associated with a higher risk of childhood overweight. Reducing the number of cigarettes during pregnancy without quitting may be beneficial for the risk of SGA but does not influence the risks of preterm birth and childhood overweight. Paternal smoking seems to be associated, independently of maternal smoking, with the risks of childhood overweight. Population strategies should focus on parental smoking prevention before or at the start of, rather than during, pregnancy.

## Supporting information

**S1 PRISMA Checklist PRISMA, Preferred Reporting Items for Systematic Reviews and Meta-Analyses.**
(PDF)

**S1 Text Protocol for analysis.**
(PDF)

**S2 Text Acknowledgments and funding.**
(PDF)

**S1 Fig. Maternal continued smoking with risks of small size for gestational age assessed by two-stage random-effects models.**
(PDF)

**S1 Table. Cohort-specific methods of data collection for parental smoking, birth outcomes, and childhood BMI. BMI, body mass index.**
(PDF)

**S2 Table. Cohort-specific description of available covariates.**
(PDF)

**S3 Table. Cohort-specific description of maternal smoking variables.**
(PDF)

**S4 Table. Associations of maternal smoking with gestational age at birth, birth weight, and childhood BMI.** BMI, body mass index.
(PDF)

**S5 Table. Change in maternal smoking habits during pregnancy, gestational age at birth, birth weight, and childhood BMI.** BMI, body mass index.
(PDF)

**S6 Table. Associations of maternal and paternal smoking with gestational age at birth, birth weight, and childhood BMI.** BMI, body mass index.
(PDF)

**S7 Table. Complete cases analysis of maternal smoking with risks of birth complications and childhood overweight (with maternal age and BMI in categories).** BMI, body mass index.
(PDF)

**S8 Table. Complete cases analysis of maternal smoking with risks of birth complications and childhood overweight (with maternal age and BMI continuously).** BMI, body mass index.
(PDF)

**S9 Table. Contact information for data requests per cohort.**
(PDF)

## Author Contributions

**Conceptualization:** Elise M. Philips, Susana Santos, Romy Gaillard, Vincent W. V. Jaddoe.

**Data curation:** Elise M. Philips, Susana Santos, Romy Gaillard.

**Formal analysis:** Elise M. Philips.

**Methodology:** Elise M. Philips, Susana Santos, Leonardo Trasande, Juan J. Aurrekoetxea, Henrique Barros, Andrea von Berg, Anna Bergström, Philippa K. Bird, Sonia Brescianini, Carol Ní Chaoimh, Marie-Aline Charles, Leda Chatzi, Cécile Chevrier, George P. Chrousos, Nathalie Costet, Rachel Criswell, Sarah Crozier, Merete Eggesbø, Maria Pia Fantini, Sara Farchi, Francesco Forastiere, Marleen M. H. J. van Gelder, Vagelis Georgiu, Keith M. Godfrey, Davide Gori, Wojciech Hanke, Barbara Heude, Daniel Hryhorczuk, Carmen Iñiguez, Hazel Inskip, Anne M. Karvonen, Louise C. Kenny, Inger Kull, Debbie A. Lawlor, Irina Lehmann, Per Magnus, Yannis Manios, Erik Melén, Monique Mommers, Camilla S. Morgen, George Moschonis, Deirdre Murray, Ellen A. Nohr, Anne-Marie Nybo Andersen, Emily Oken, Adriëtte J. J. M. Oostvogels, Eleni Papadopoulou, Juha Pekkanen, Costanza Pizzi, Kinga Polanska, Daniela Porta, Lorenzo Richiardi, Sheryl L. Rifas-Shiman, Nel Roeleveld, Franca Rusconi, Ana C. Santos, Thorkild I. A. Sørensen, Marie Standl, Camilla Stoltenberg, Jordi Sunyer, Elisabeth Thiering, Carel Thijs, Maties Torrent, Tanja G. M. Vrijkotte, John Wright, Oleksandr Zvinchuk, Romy Gaillard, Vincent W. V. Jaddoe.

**Supervision:** Vincent W. V. Jaddoe.

**Writing – original draft:** Elise M. Philips, Susana Santos, Vincent W. V. Jaddoe.

**Writing – review & editing:** Leonardo Trasande, Juan J. Aurrekoetxea, Henrique Barros, Andrea von Berg, Anna Bergström, Philippa K. Bird, Sonia Brescianini, Carol Ní Chaoimh, Marie-Aline Charles, Leda Chatzi, Cécile Chevrier, George P. Chrousos, Nathalie Costet, Rachel Criswell, Sarah Crozier, Merete Eggesbø, Maria Pia Fantini, Sara Farchi, Francesco Forastiere, Marleen M. H. J. van Gelder, Vagelis Georgiu, Keith M. Godfrey, Davide Gori, Wojciech Hanke, Barbara Heude, Daniel Hryhorczuk, Carmen Iñiguez, Hazel Inskip, Anne M. Karvonen, Louise C. Kenny, Inger Kull, Debbie A. Lawlor, Irina Lehmann, Per Magnus, Yannis Manios, Erik Melén, Monique Mommers, Camilla S. Morgen, George Moschonis, Deirdre Murray, Ellen A. Nohr, Anne-Marie Nybo Andersen, Emily Oken, Adriëtte J. J. M. Oostvogels, Eleni Papadopoulou, Juha Pekkanen, Costanza Pizzi, Kinga Polanska, Daniela Porta, Lorenzo Richiardi, Sheryl L. Rifas-Shiman, Nel Roeleveld, Franca Rusconi, Ana C. Santos, Thorkild I. A. Sørensen, Marie Standl, Camilla Stoltenberg, Jordi Sunyer, Elisabeth Thiering, Carel Thijs, Maties Torrent, Tanja G. M. Vrijkotte, John Wright, Oleksandr Zvinchuk, Romy Gaillard.

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
