## [Editor Report · Decision Letter 0]

2 Jan 2020

Dear Dr Jaddoe, 

Thank you for submitting your manuscript entitled "Changes in parental smoking during pregnancy and risks of adverse birth outcomes and childhood overweight: an individual participant data meta-analysis of 230,000 families" for consideration by PLOS Medicine.

Your manuscript has now been evaluated by the PLOS Medicine editorial staff and I am writing to let you know that we would like to send your submission out for external peer review.

Kind regards,

Louise Gaynor-Brook, MBBS PhD

Associate Editor, PLOS Medicine

---

## [Decision Letter · Decision Letter 1]

2 Mar 2020

Dear Dr. Jaddoe,

Thank you very much for submitting your manuscript "Changes in parental smoking during pregnancy and risks of adverse birth outcomes and childhood overweight: an individual participant data meta-analysis of 230,000 families" (PMEDICINE-D-19-04667R1) for consideration at PLOS Medicine. 

Your paper was evaluated by a senior editor and discussed among the editors here. It was also discussed with an academic editor with relevant expertise, and sent to independent reviewers, including a statistical reviewer. The reviews are appended at the bottom of this email and any accompanying reviewer attachments can be seen via the link below:

[LINK]

In light of these reviews, I am afraid that we will not be able to accept the manuscript for publication in the journal in its current form, but we would like to consider a revised version that addresses the reviewers' and editors' comments. Obviously we cannot make any decision about publication until we have seen the revised manuscript and your response, and we plan to seek re-review by one or more of the reviewers. 

We expect to receive your revised manuscript by Mar 23 2020 11:59PM. Please email us (plosmedicine@plos.org) if you have any questions or concerns.

We look forward to receiving your revised manuscript. 

Sincerely,

Louise Gaynor-Brook, MBBS PhD

Associate Editor 

PLOS Medicine

plosmedicine.org

General comment: Please remove all language that implies causality, throughout your manuscript. Reference should be made to associations instead.

General comment: Please cite reference numbers in square brackets, leaving a space before the reference bracket, and removing spaces between reference numbers where more than one reference is cited. 

General comment: Please rename supplementary figures/tables as Supplementary Figure / Table, etc rather than eFigure and eTable

Please revise your title according to PLOS Medicine's style, placing the study design in the subtitle (ie, after a colon). We suggest “Changes in parental smoking during pregnancy and risks of adverse birth outcomes and childhood overweight in Europe and North America: an individual participant data meta-analysis of 229,000 singleton births” or similar.

Data Availability Statement: PLOS Medicine requires that the de-identified data underlying the specific results in a published article be made available, without restrictions on access, in a public repository or as Supporting Information at the time of article publication, provided it is legal and ethical to do so. Please see the policy at http://journals.plos.org/plosmedicine/s/data-availability and FAQs at 

http://journals.plos.org/plosmedicine/s/data-availability#loc-faqs-for-data-policy

Please provide appropriate contact(s) (web or email address) to whom requests for access to de-identified data can be made. Please note that this cannot be a study author. 

Please remove the ‘Research in context’ section, replacing instead with a short, non-technical Author Summary of your research to make findings accessible to a wide audience that includes both scientists and non-scientists. The Author Summary should immediately follow the Abstract in your revised manuscript. This text is subject to editorial change and should use non-identical language, that is distinct from the scientific abstract. Please see our author guidelines for more information: https://journals.plos.org/plosmedicine/s/revising-your-manuscript#loc-author-summary

Please report your abstract according to PRISMA for abstracts, following the PLOS Medicine abstract structure (Background, Methods and Findings, Conclusions) http://www.plosmedicine.org/article/info:doi/10.1371/journal.pmed.1001419

Abstract Background: Please expand upon the context of why the study is important. The final sentence should clearly state the study question.

Please combine the Methods and Findings components of your Abstract under one subheading of ‘Methods and Findings’.

Please include brief demographic details of the populations included in the meta-analysis (e.g. age ranges, nationalities, parity of women, etc.), years during which the studies took place, and further details of the study settings (i.e. which countries in Europe and North America; from where were women recruited e.g. hospitals, community settings, etc.)

It is not clear how many studies include parental smoking and how many are just maternal or paternal - please mention this in the abstract

Please include the important dependent variables that are adjusted for in the analyses.

There is no need to define OR and CI in the abstract as these are standard abbreviations.

In the last sentence of the Abstract Methods and Findings section, please describe the main limitation(s) of the study's methodology.

Lines 41-44 - Please quantify the results presented with OR and 95% CI

Please replace ‘Interpretation’ with the ‘Conclusions’

Please begin this section with "In this study, we observed ..." or similar. Please address the study implications, emphasizing what is new without overstating your conclusions.

Introduction

Please expand your Introduction to outline past research and explain the need for and potential importance of your study. Indicate whether your study is novel and how you determined that. If there has been a systematic review of the evidence related to your study (or you have conducted one), please refer to and reference that review and indicate whether it supports the need for your study. 

Line 59 - please provide the range for % women who smoke during pregnancy, or revise the term ‘range’

Line 61 - Please revise or remove sentence "Also, since paternal and maternal smoking often cluster within families, insight into the effects of combined parental smoking may help to improve family-focused prevention strategies", as prevention strategies are not included in the scope of this study.

Methods

Please incorporate eFigure 1 into the main text of your manuscript.

Please add the following statement, or similar, to the Methods: "This study is reported as per the Preferred Reporting Items for Systematic Reviews and Meta-Analyses (PRISMA) guideline (S1 Checklist)."

When completing the PRISMA checklist, please use section and paragraph numbers, rather than page numbers.

Please complete your PRISMA checklist - several boxes not ticked. We note that some form of search strategy was used (even if not like a conventional meta-analysis) - please elaborate on how cohorts were identified (line 75 - "cohort studies were invited") and how eligibility for inclusion in the meta-analysis was decided? 

Please expand upon the methods used to check quality and heterogeneity, which are paramount for meta-analyses.

We note that all covariates are categorised. Please ensure that certain variables such as age not categorised (please refer to report from Reviewer 1)

Please confirm that written or oral informed consent was obtained in all cohort studies included.

Results

Lines 161-7 - Please quantify the results presented with OR and 95% CI

All tables - please define all abbreviations used in the table legend (except cohort names). 

Please incorporate supplementary figures into the main text. 

Discussion

Please remove all subheadings throughout your discussion.

Please present and organize the Discussion as follows: a short, clear summary of the article's findings; what the study adds to existing research and where and why the results may differ from previous research; strengths and limitations of the study; implications and next steps for research, clinical practice, and/or public policy; one-paragraph conclusion.

Requests from the academic editor:

The authors imply causation in several places - eg "Quitting smoking in the first trimester place mothers at the same risk of preterm birth and small size for gestational age as non-smoking mothers."

I think the language should be tempered throughout to make it clear that the findings are based on observational data and associations can be seen, but causation cannot be assessed. The authors do discuss, that "shared family-based lifestyle and genetic characteristics are potential underlying mechanisms" for childhood overweight; and this makes sense. However, the abstract implies that maternal and paternal smoking are directly linked to childhood overweight.

Comments from the reviewers:

Reviewer #1: See attachment

Michael Dewey

Reviewer #2: This study stems from the knowledge that prenatal exposure to maternal smoking carries risks for the baby and it investigates the important question of whether quitting smoking during early pregnancy reduces the risks of birth complications and childhood obesity, compared to continued smoking. The authors have conducted a powerful study by meta-analysing data from 28 European cohorts reaching an impressive N=229158 families. They conclude that quitting early in pregnancy is beneficial in reducing the risk of low birth weight. This study is commendable since it is powerful and addresses an important and needed question with immediate clinical relevance. However, before its publication I would like the authors to carry out revisions according to the following points:

Main considerations:

1) The authors mention in the Introduction that smoking leads to, amongst other outcomes, congenital abnormalities, still birth and sudden infant death syndrome, which are major complications, but the study focuses on pre-term birth, low birth weight and childhood overweight, potentially less life-threatening. What was the reason to leave out the most detrimental outcomes from this study? If this study is to be translated to a clinical application it would need to be relevant to families in reducing risks of major complications. Perhaps the authors could emphasised how detrimental pre-term births, low birth weight and being overweight in childhood are for long-term health to add context to this research. 

2) The Methods (either main text or supplemental) section needs an explanation per cohort on how smoking and the covariates were measured and derived. It is very unlikely that the type of information is the same across all these numerous studies. For instance, was the question asked to the mothers whether they smoked in the first trimester and similarly for the later time-points or was the question whether they smoked in the last two weeks or currently relative to a questionnaire? Some studies contain many variables related to smoking, so it is important to know what was used to create the smoking variables used here. Even if questionnaires are aimed at first trimester, depending on the study they might have been returned at various times during pregnancy. Some women might have been enrolled late in the study etc. so the accuracy of the time of this self-reported information in relation to the time in pregnancy can vary and it is important to specify all this information to understand the results and to be able to reproduce the study. Particularly as some cohorts are much bigger in size than others and they might have affected the results more.

3) At lines 136-137: What was the reason for categorising the missing participants as an additional group rather than conducting multiple imputation? This approach could add bias in the estimates as shown for instance by Groenfeld et al (CMAJ. 2012 Aug 7; 184(11): 1265-1269)?

4) The authors have not commented on the limitation that for some of the analyses the sample size was very small (for instance in the dose-stratified analyses in Table 3) and when they compare the effect of a change in habit between first-trimester-only smoking to continued or the effect of a dose change they are effectively comparing the different associations (odds ratios). However, differences in samples sizes lead to differences in power and therefore the associations are not always comparable. They should make conclusions only regarding the associations for which there is enough evidence to conclude that there is a risk, for instance the effect of continued smoking (based on N~5000) on small for gestational age. They cannot rule out an effect of smoking only in the first trimester (based on N~200). 

5) Some of the effects considered have confidence interval very close to 1, so the authors should be more cautious in concluding a risk since even if there was an effect it might be small and not necessarily meaningful. 

6) Could the effect of paternal smoking on 'childhood overweight' suggest that there is some residual confounding? 

7) The authors should state clearly in the abstract, discussion and conclusions that this is an observational study, rather than causal, and the difference in associations between different smoking behaviours might be due to familial characteristics rather than different exposure to smoke, which could be measured by cotinine levels for instance, at different timepoints in pregnancy in relation to birth outcomes. 

8) Could the authors investigate more the issue of high heterogeneity in the small for gestational age analysis by conducting some sensitivity analysis? For instance they could conduct the analysis only including the cohorts that have both data (first trimester and continued) and compare the associations found across these to check that they remain similar to what they have found. The authors should consider the effect that excluding large cohorts from the first trimester data could have on the continued smoking analyses.

Other points:

9) At lines 104-105: what category was used as reference for gestational age? The authors mentioned the definition used for SGA and that "appropriate and large size for gestational age were used as the reference group". Do they mean that everyone else not in the category for SGA was the reference group?

10) At lines 159-160: Could the authors add info on heterogeneity in the other models too? Could they also add in the Discussion what could have contributed to the high heterogeneity for the small for gestational age outcome?

Reviewer #3: This is an individual participant data meta-analysis of prospective cohort studies to examine the associations of prenatal parental smoking with adverse birth outcomes and childhood overweight. It is a well-described manuscript, with large study sample. It would be much better if the authors more discuss the biological mechanisms in the discussion session. For example, the authors referred to cadmium and DNA methylation, which is very interesting, but do they explain the different results in the associations of the outcomes?

[LINK]

---

## [Decision Letter · Decision Letter 2]

21 May 2020

Dear Dr. Jaddoe,

Thank you very much for re-submitting your manuscript "Changes in parental smoking during pregnancy and risks of adverse birth outcomes and childhood overweight in Europe and North America: an individual participant data meta-analysis of 229,000 singleton births" (PMEDICINE-D-19-04667R2) for consideration at PLOS Medicine.

I have discussed the paper with our academic editor and it was also seen again by three reviewers. I am pleased to tell you that, provided the remaining editorial and production issues are fully dealt with, we expect to be able to accept the paper for publication in the journal.

[LINK]

Please let me know if you have any questions. Otherwise, we look forward to receiving the revised manuscript shortly. 

Sincerely,

Richard Turner, PhD

rturner@plos.org

Requests from Editors:

Please confirm whether or not all authors have agreed to the removal of Michelle Taylor as an author. 

At line 20, for example, please avoid "effect" (estimate) given the research design, in favour of "risk estimate", say.

In your abstract and throughout the paper, please quote p values alongside 95% CI, where available. 

Early in the methods section of the main text, please state whether or not the study had a protocol or prespecified analysis plan (and if so attach the document as a supplementary file, referred to in the text). Please highlight analyses that were not prespecified. 

At line 143, would that be written informed consent?

Please add "In this study ..." or similar to begin the sentence at line 380.

Please adapt your text around line 470 to signpost the discussion of study limitations, e.g., by adding "among study limitations ...".

At line 472, please adapt the text to "our study might have been underpowered" or similar. 

Please read through the text and adapt punctuation where necessary; for example, at line 33 "...at the start, rather than during, pregnancy." would seem more readable.

Competing interest information - currently at the end of the ms - should appear only in the metadata (via the submission form). 

Please move the "details of ethics approval" statement at the end of the ms to the methods section of the main text, and state whether consent was informed. 

Please move the lengthy acknowledgements and funding information at the end of the main text to a supplementary file. 

As for the rest of the article, please adapt the figures so that p values are quoted as "p<0.001" where appropriate.

Comments from Reviewers:

*** Reviewer #1: 

The authors have addressed all my points.

Michael Dewey

*** Reviewer #2: 

The authors have done a good job in addressing the comments. Please consider these corrections too:

Abstract, line 20: what effect sizes are smaller than what? As it reads it does not look like 1.89, 1.93 and 2.79 are that much smaller compared to 2.15, with overlapping confidence intervals. The authors should rephrase these results in a more easily interpretable way.

Line 121: usually maternal smoking is assumed to affect the foetus because of in-utero effects. What was the reason for choosing also paternal smoking? Initially I thought it was a negative control, i.e. if effects of paternal are similar to maternal ones there is less causal evidence as these could be attributable to shared confounding such as low socioeconomic status. Could the authors state more clearly the rationale for choosing to investigate paternal smoking too and discussing the results in the discussion in view of their initial hypothesis.

*** Reviewer #3:

[no further comments]

***

[LINK]

---

## [Editor Report · Decision Letter 3]

9 Jul 2020

Dear Dr. Jaddoe, 

On behalf of my colleagues and the academic editor, Dr. Sarah J Stock, I am delighted to inform you that your manuscript entitled "Changes in parental smoking during pregnancy and risks of adverse birth outcomes and childhood overweight in Europe and North America: an individual participant data meta-analysis of 229,000 singleton births" (PMEDICINE-D-19-04667R3) has been accepted for publication in PLOS Medicine. 

PRODUCTION PROCESS

PRESS

PROFILE INFORMATION

Thank you again for submitting the manuscript to PLOS Medicine. We look forward to publishing it. 

Best wishes, 

Richard Turner, PhD

Senior Editor 

PLOS Medicine

plosmedicine.org